# Field control of quasiparticle decay in a quantum antiferromagnet

Shunsuke Hasegawa [1], Hodaka Kikuchi [1], Shinichiro Asai[1], Zijun Wei[1], Barry Winn[2], Gabriele Sala [2], Shinichi Itoh [3] & Takatsugu Masuda [1,3,4] ✉

Dynamics in a quantum material is described by quantized collective motion: a quasiparticle. The single-quasiparticle description is useful for a basic understanding of the system, whereas a phenomenon beyond the simple description such as quasiparticle decay which affects the current carried by the quasiparticle is an intriguing topic. The instability of the quasiparticle is phenomenologically determined by the magnitude of the repulsive interaction between a single quasiparticle and the two-quasiparticle continuum. Although the phenomenon has been studied in several materials, thermodynamic tuning of the quasiparticle decay in a single material has not yet been investigated. Here we show, by using neutron scattering, magnetic field control of the magnon decay in a quantum antiferromagnet $RbFeCl_3$, where the interaction between the magnon and continuum is tuned by the field. At low fields where the interaction is small, the single magnon decay process is observed. In contrast, at high fields where the interaction exceeds a critical magnitude, the magnon is pushed downwards in energy and its lifetime increases. Our study demonstrates that field control of quasiparticle decay is possible in the system where the two-quasiparticle continuum covers wide momentum-energy space, and the phenomenon of the magnon avoiding decay is ubiquitous.

The concept of a quasiparticle has been successful in explaining various types of low-energy excitations, including charge, spin, and lattice, in many-body systems. Using a spectroscopic approach, a weakly coupled quasiparticle with a long lifetime can be probed as a well-defined excitation, allowing identification of the effective Hamiltonian and basic understanding of the system. Momentum-resolved spectroscopy has permitted investigations into the intricate structure of spectra, revealing the effect of quasiparticle interactions that results in the renormalization of the dispersion[1–3] and instability of the quasiparticle[4–8].

The microscopic phenomena in the spectra affect the bulk properties of materials. In the thermoelectric material PbTe, the interaction between longitudinal acoustic and transverse optical modes (here, the quasiparticles are phonons) induces the decay and overdamping of the former phonon in the low energy region, leading to the low conductivity of thermal current[2]. The instability of the quasiparticle is key for the current to exist in the bulk property.

Two examples illustrate that the instability of the quasiparticle is changed by the interaction between the one-quasiparticle and two-quasiparticle continuum[9]. An example of a case of the strong interaction is found in the longitudinal sound wave, phonon, in superfluid $^4$He[4,6,10]. The spectrum in low energy exhibits a local minimum with energy $\Delta$, called a roton, for which qualitative behavior is explained by Feynman and Cohen's (FC) harmonic dispersion[11]. However, the spectrum does not exceed a critical energy of $2\Delta$, which is the lower boundary of the two-phonon continuum. The strong interaction between the one-phonon and continuum pushes one-phonon energy downwards, and the one-phonon stays at $2\Delta$ outside the continuum. On the FC dispersion beyond the critical energy, the bare one-phonon decays into a pair of protons, and a remnant of one-phonon, which is

[1]Institute for Solid State Physics, The University of Tokyo, Chiba 277-8581, Japan. [2]Neutron Scattering Division, Oak Ridge National Laboratory, Oak Ridge, TN 37831, USA. [3]Institute of Materials Structure Science, High Energy Accelerator Research Organization, Ibaraki 305-0801, Japan. [4]Trans-scale Quantum Science Institute, The University of Tokyo, Tokyo 113-0033, Japan. ✉e-mail: masuda@issp.u-tokyo.ac.jp

ascribed to the bound state of two-phonons, was observed. The phenomenon is considered universal in bosonic systems and has also been observed in a spin-gap antiferromagnet $BiCu_2PO_6$[8].

An example of a weak interaction is found in a two-dimensional quantum magnet, piperazinium hexachlorodicuprate (PHCC)[6]. This case is simple; the quasiparticle decays in the continuum, and a remnant one-magnon is probed as a broad excitation. The conjecture that this work tests is that if one tunes the interaction between a quasiparticle and the continuum in an identical material by applying an external field, would the quasiparticle decay behavior change?

This study examines magnon decay in a triangular lattice quantum antiferromagnet $RbFeCl_3$. The magnetism of $Fe^{2+}$ ion surrounded by $Cl^-$ octahedra with trigonal distortion is effectively described by an $S = 1$ spin with strong easy-plane anisotropy[12]. $Fe^{2+}$ ions form a one-dimensional ferromagnetic chain along the crystallographic $c$-axis, and the interchain interaction in the triangular lattice in the $ab$-plane is antiferromagnetic[13] (see Fig. S2 in Supplementary Information for the crystal structure). At low temperatures, the compound exhibits a non-collinear 120° structure due to the frustration[14,15]. The spectrum was qualitatively similar to that of the pressure-induced ordered state in the isostructural compound $CsFeCl_3$ near the quantum critical point (QCP)[3,16], which cannot be explained by standard linear spin wave theory[17]. Instead, the strong hybridization of the transverse and longitudinal fluctuations resulting from the non-collinear magnetic structure renormalizes the magnetic excitation, as explained by the linear extended spin wave theory (LESW)[3,18], which is also known as a generalized SU(3) spin-wave theory[19], as well as (1+1)-dimensional quantum field theory[20,21] and Lagrangian spin-wave theory[22]. Because a non-colinear magnetic structure is realized near the QCP and the excitation is strongly hybridized with longitudinal fluctuation, the magnon decay[23–29] is anticipated in wide four-dimensional momentum-energy space. Furthermore, gapless behavior and a large dispersion perpendicular to the triangular lattice yield a two-magnon continuum covering the whole region of one-magnon excitation.

In this study, we performed inelastic neutron scattering (INS) measurements in the magnetic field on $RbFeCl_3$ to study the magnon decay and the interaction between single magnons and the two-magnon continuum. We observed a simple magnon decay in a low field

where the interaction is small and a magnon avoiding decay in a high field where the interaction was large. Thus, we succeeded in controlling the magnon decay using the field. In contrast with the avoided phonon decay in superfluid $^4He$ and magnons in magnetic materials previously reported[8,9], the phenomenon was observed in the presence of a two-quasiparticle continuum, indicating that the phenomenon is not limited to outside the continuum but also occurs inside.

## Results

### Inelastic neutron scattering spectra

The measured spectra along high symmetry directions of the momentum transfer **q** at the zero field are shown in Fig. 1a, and high symmetry points in the reciprocal space are shown in Fig. 1b, c. The dispersive excitations with bandwidths of 3.5 meV along the $c^*$ direction and 1.5 meV in the $a^*$–$b^*$ plane were observed. The energy maximum of the dispersion along the $c^*$ direction at $l = 1$ corresponded to the ferromagnetic interaction between the Fe spins aligned with the $c/2$ spacing. At K points with $l = 0$, the gapless Nambu–Goldstone and gapped hybridized modes were observed, which were consistent with previous INS experiments[17,30,31]. The dispersion relations were well reproduced using LESW, as shown by the curves, which are described in Supplementary Information Section IIA.

The magnetic field ($H$) dependence of INS spectra along **q** = $(2h, -h + 0.5, 0)$ is shown in Fig. 2a–f. The field was applied along the $c$-axis. Due to Zeeman splitting, two gapped excitations in addition to the gapless Nambu-Goldstone mode were observed at 1 T. With the increase in the field, the Zeeman split was more pronounced, and at $H \geq 3$ T, the spectra were divided into the low-energy band at $\hbar\omega \lesssim 1$ meV and the high-energy band at $\hbar\omega \gtrsim 1.2$ meV. Simultaneously, another mode appeared in the low-energy band, which had a local maximum at K points and a local minimum at Γ point. These features were semiquantitatively reproduced using LESW, as shown in Fig. 2g–l, though the discrepancy will be pointed out and discussed later.

We observed the line broadenings of magnons around Γ point from 0 to 4 T, as indicated by the red arrows in Fig. 2a–e, and around the K points from 3 to 5 T in Fig. 2d–f. The highest energy mode at Γ point got blurred with an increase of the field, but at $H = 4$ T, it became well-defined, as indicated by the white arrows in Fig. 2e. This means that the lifetime of the magnon is changed by the field. The energies of the well-defined magnons in Fig. 2e, f were lower than those of the calculated ones in Fig. 2k, l. To corroborate the change in the spectrum, the field dependences of the constant **q** cuts at K and Γ points are shown in Fig. 3a, b, respectively. We fitted the data using the Voigt function indicated by the solid curves. The estimated peak energies are shown by symbols in Fig. 3c, d, and those for the full-width at half maximum of the Lorentzian part ($FWHM_L$), which represents the intrinsic line broadening of the magnon, corrected for instrumental energy resolution, are shown in Fig. 3e, f. The method for estimating the instrumental resolution is described in the method section and in Supplementary Information Section III. The observed excitations were categorized according to the calculated mode, with the group of $\hbar\omega_1$ in the low energy region indicated by blue symbols, the group of $\hbar\omega_2$ in the high energy region by red symbols, and the other excitations by gray symbols in Fig. 3c–f. The peaks belonging to the group of $\hbar\omega_1$ were resolution-limited, and the $FWHM_L$ was zero. In contrast, those belonging to the group of $\hbar\omega_2$ showed finite $FWHM_L$, which varied with the wave vector and magnetic field.

### Nontrivial spectrum split

The excitation energies calculated using LESW at K and Γ points are shown by solid curves in Fig. 3c, d, respectively. The experimental data at K point indicated by symbols were reasonably reproduced by calculation (see Fig. S1 in Supplementary Information Section IB for the data at M point, which was also reproduced by the calculation), but those at Γ point, particularly red squares, deviated from the calculation at $H \geq 3$ T.

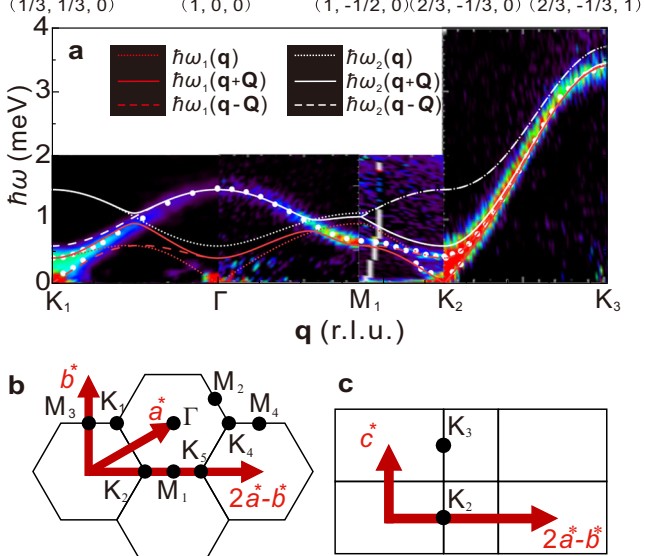

**Fig. 1 | Inelastic neutron scattering (INS) spectra measured at zero field. a** False color map of INS spectra acquired at HRC. White circles represent peak energy obtained from fitting results of constant **q** cuts. Dotted, solid, and dashed curves are dispersion relations calculated using linear extended spin wave theory using best-fit parameters. **b, c** High-symmetry points in reciprocal space.

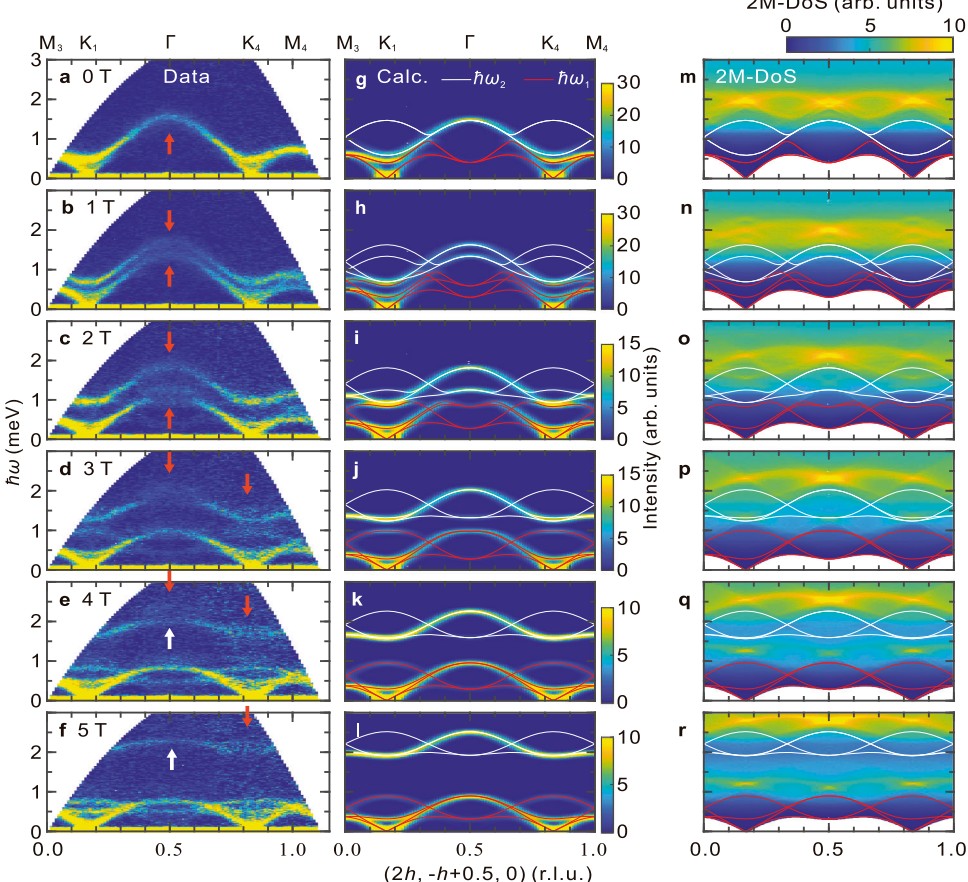

**Fig. 2 | False color map of inelastic neutron scattering (INS) spectra and calculated two-magnon density of state in RbFeCl$_3$. a–l** Magnetic field ($H \| c$-axis) dependences of false color maps for observed (**a–f**) and calculated (**g–l**) INS spectra. Red arrows in **a–f** indicate broadening linewidth. White arrows in **e, f** indicate magnons avoiding decay. White and red solid curves in **g–l** are one-magnon dispersion relations of $\hbar\omega_1$ and $\hbar\omega_2$ modes calculated using linear extended spin wave theory (LESW) using the best-fit parameters. The calculated spectra are convoluted by the instrumental resolution. **m–r** Two-magnon density of state (2M-DoS) calculated from one-magnon dispersion relations using LESW.

In the field region in Fig. 3b, a nontrivial spectrum split indicated by red and gray arrows was observed. This split was not explained by the calculation. The linewidths of the peaks indicated by the red arrows at 4 and 5 T were sharply compared with that of the highest energy peak at 2 T, which is also shown by the red squares in Fig. 3f. The behavior was observed as the well-defined spectrum indicated by the white arrows in Fig. 2e, f as well. The linewidth indicated by the gray arrow was broadened with the field, and it became a continuum-like excitation at 5 T. Because an excitation with a small linewidth is regarded as an approximate eigenstate of a single-quasiparticle Hamiltonian, the former was named $\hbar\omega_2$ mode, and the latter was named remnant magnon RM$_H$, despite the peak energy of $\hbar\omega_2$ mode being pushed below the calculated energy and that of RM$_H$ coincides with the energy as shown in Fig. 3d. The nontrivial spectrum split at $H \geq 3$ T is the magnon analog of the quasiparticle avoiding decay reported in superfluid $^4$He[4,6,10], which was observed in a spin-gap antiferromagnet BiCu$_2$PO$_6$ at zero magnetic field as well[8]. In contrast, the simple broad spectra at $H \leq 2$ T corresponds to a magnon decay reported in PHCC[6].

In the low energy range near the $\hbar\omega_1$ mode, continuum-like excitations, named RM$_L$ were observed, indicated by light gray arrows in Fig. 3b. The maximum of RM$_L$ and the peak energy of $\hbar\omega_1$ were close to one another compared with the case in $\hbar\omega_2$ mode.

## Discussion
### Magnon decay and two-magnon density of state
Because the ground state of RbFeCl$_3$ is magnetically ordered with a non-collinear structure near the QCP, the excitation exhibits both

transverse and longitudinal fluctuations of spins in each mode, unlike semiclassical non-collinear magnets[23,24] or quantum collinear magnets[25–29]. Our LESW calculation elucidated that $\hbar\omega_1$ modes are dominated by transverse fluctuation, whereas $\hbar\omega_2$ modes have significant longitudinal fluctuation (Supplementary Information IIB). The broadening of the linewidth observed in the energy region of $\hbar\omega_2$ modes suggested that longitudinal magnon decay dominates the decay origin.

For the magnon to decay into two magnons, the kinematics must satisfy momentum and energy conservation[32–35]: $\hbar\omega(\mathbf{q}) = \hbar\omega(\mathbf{q}_1) + \hbar\omega(\mathbf{q}_2)$ and $\mathbf{q} = \mathbf{q}_1 + \mathbf{q}_2$, where $\hbar\omega(\mathbf{q})$ is the energy of a one-magnon with momentum $\mathbf{q}$. The region of the decay channel in the momentum-energy ($\mathbf{q}$-$\hbar\omega$) space corresponds to that for the two-magnon continuum; $\hbar\omega_{2\text{mag}}(\mathbf{q}) = \sum_{\mathbf{q}_1} \hbar\omega(\mathbf{q}_1) + \hbar\omega(\mathbf{q} - \mathbf{q}_1)$. Because the density of state of the two-magnon (2M-DoS) is the number of the magnon decay channel, it is a good indicator of the decay rate. The calculated 2M-DoS using the one-magnon energy obtained using LESW is presented in Fig. 2m–r (see Supplementary Information IIC). The 2M-DoS covers the whole region of one-magnon modes except the lowest energy mode in all magnetic fields because of the large dispersion perpendicular to the triangular lattice and the gapless feature at K points. We found that the 2M-DoS was large in the $\hbar\omega_2$ modes region and small in that of $\hbar\omega_1$ modes. This is consistent with the observation that magnon decay is only detected in $\hbar\omega_2$ modes. At the Γ point, the 2M-DoS had its maximum near the active $\hbar\omega_2$ in all fields, which explains why the line broadening was enhanced there. At the K points, the $\hbar\omega_2$ mode was separated from the $\hbar\omega_1$ mode at $H \geq 3$ T with the line

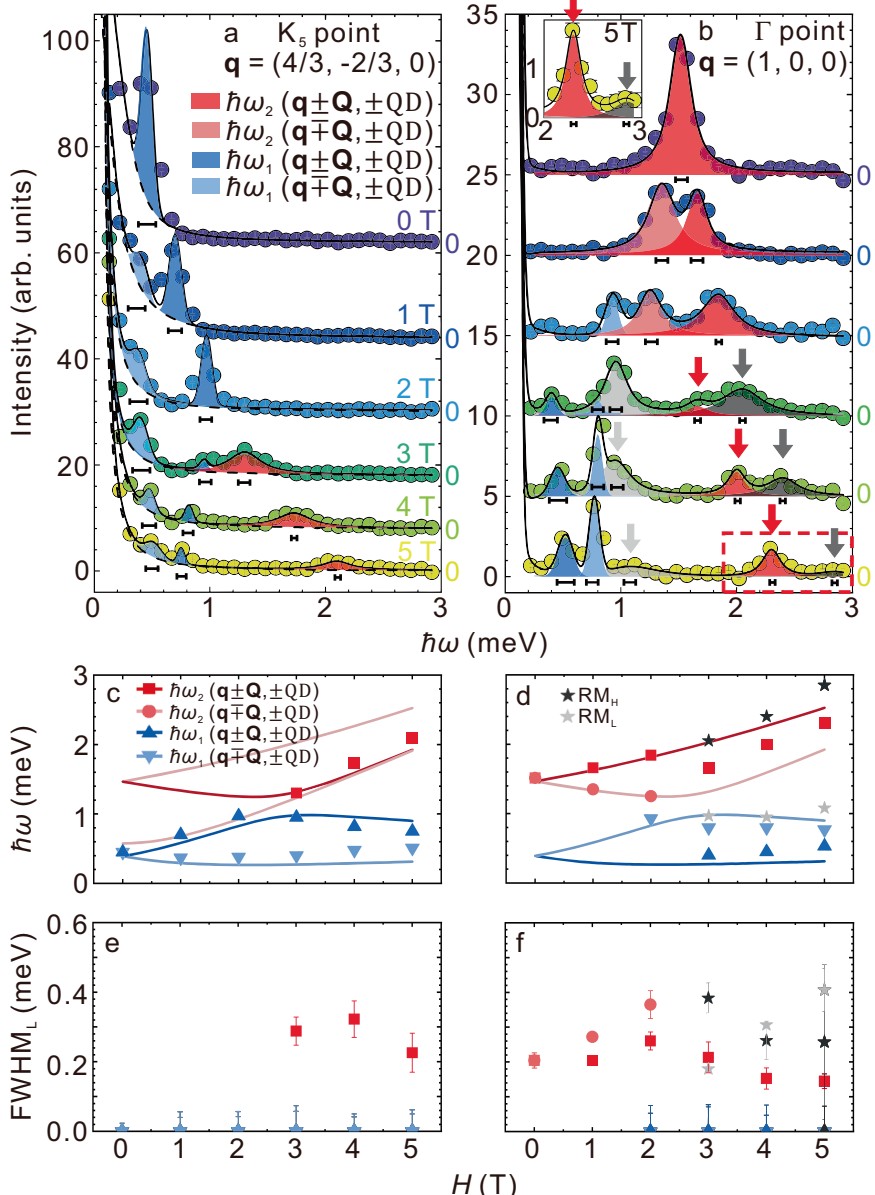

**Fig. 3 | Constant q cuts, energies of excitations, and line widths. a, b** Magnetic field dependences of constant **q** cuts at K (−4/3, 2/3, 0) and Γ (1, 0, 0). The peaks indicated by deep and light blue are $\hbar\omega_1$ modes, and those by deep and light red are $\hbar\omega_2$ modes. The horizontal bars represent the instrumental resolution. The peaks indicated by deep and light gray are remnant magnons $RM_H$ and $RM_L$. Error bars represent Poissonian noise. The inset shows enlarged data in the high-energy region at 5 T, indicated by a red dashed square. Red and gray arrows indicate a nontrivial spectrum split. **c, d**: Magnetic field dependence of peak energies at $K_5$ point in **c** and at Γ point in **d**. Symbols and solid curves indicate experimental data and calculation by linear extended spin wave theory. Deep and light red colors are $\hbar\omega_2$ modes, and deep and light blue colors are $\hbar\omega_1$ modes. See Supplementary Information IIB and Fig. S3 for the details of the modes. **e, f** Magnetic field dependence of the intrinsic linewidth $FWHM_L$. Error bars for $\hbar\omega_2$ modes and remnant magnons $RM_H$ and $RM_L$ represent fitting uncertainties. Error bars for $\hbar\omega_1$ are the instrumental resolutions.

broadening, and the energy was close to the local maximum of 2M-DoS at 3 T.

## Magnetic field control of magnon decay

The repulsive interaction between the one-magnon and continuum is enhanced (suppressed) at a large (small) 2M-DoS. We, thus, assumed the magnitude of repulsive interaction as $RI = D_{2M}^{max}/(E_{2M}^{max} - \hbar\omega_i^{calc})$, where $D_{2M}^{max}$ and $E_{2M}^{max}$ are the maximum of 2M-DoS and its energy, respectively. Because the 2M-DoS and one-magnon energy are the calculated values obtained by LESW and the one-magnon energy did not reproduce the experiment perfectly, $RI$ used here is the qualitative indicator. We display $RI$ and the deviation of the observed one-magnon energy from the calculated one, $\hbar\omega_i^{obs} - \hbar\omega_i^{calc}$, for $\hbar\omega_2(\mathbf{q} \pm \mathbf{Q}, \pm QD)$

mode as a function of $H$ in Fig. 4a (see Supplementary Information IIB and Fig. S3 for the definition of the mode). The $RI$ monotonically increased with the field at $H \leq 4$ T. At low fields where $RI$ was small, the one-magnon having finite linewidth remained at the calculated energy. This case was previously observed in two-dimensional quantum magnet PHCC[6]. At approximately 3 T, $RI$ gradually increased with the field, and $\hbar\omega_i^{obs} - \hbar\omega_i^{calc}$ drastically decreased. The one-magnon calculated as located in the large 2M-DoS region was pushed downwards to the small 2M-DoS region. Hence, the magnon extends its lifetime. The remnant magnon stayed near the calculated energy of the one-magnon, as indicated by dark gray stars in Fig. 3d. This case was observed in superfluid $^4$He[4,6,10] and BiCu$_2$PO$_6$[8], which was discussed in the phenomenological theory[9]. The change in the one-magnon dispersion and

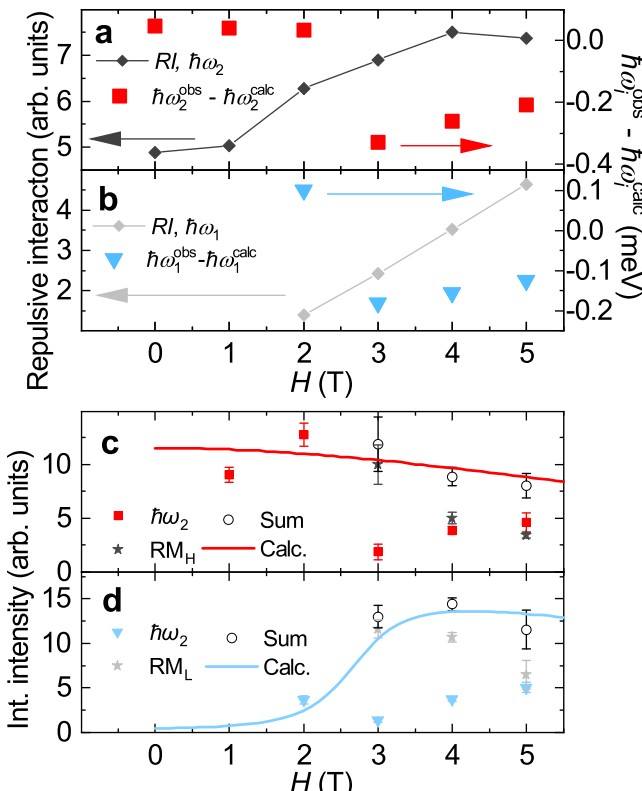

**Fig. 4 | Magnetic field dependence of repulsive interaction between one-magnon and two-magnon continuum and of integrated intensities for inelastic neutron scattering spectra. a, b** Repulsive interaction ($RI$) and the deviation of the observed one-magnon energy from the calculated one, $\hbar\omega_i^{\mathrm{obs}} - \hbar\omega_i^{\mathrm{calc}}$. Those for $\hbar\omega_2$ mode are in **a** and for $\hbar\omega_1$ mode are in (**b**). **c, d** Magnetic field dependences of integrated intensities for the one-magnon, the remnant magnon (RM$_H$ and RM$_L$), the sum of these, and calculated intensity for one-magnon by linear extended spin wave theory. Those for $\hbar\omega_2$ and RM$_H$ mode are in (**c**), and for $\hbar\omega_1$ and RM$_L$ mode are in (**d**). Error bars represent fitting uncertainties.

2M-DoS by the field tuned the interaction between the one-magnon mode and the two-magnon continuum, leading to the magnon avoiding decay at the high field. Thus, we controlled the magnon decay using the magnetic field in a single material.

The change of the spectrum from the weak interaction case to the strong interaction case is drastic in the experiment. In a phenomenological model[9], the change is cross-over like in one or two dimensions; there is always a long-lived quasiparticle, and the weight is gradually shifted from the continuum onto the separate long-lived mode. In contrast, in three dimensions, the change is phase-transition-like; there is a threshold where the long-lived mode appears. Even though RbFeCl$_3$ is weakly coupled ferromagnetic chains, the system is three-dimensional due to interchain coupling, which leads to a drastic change in the spectrum. Meanwhile, the remnant magnon at $H \geq 3$ T would be due to the one-dimensional substructure. Further theoretical study considering the specific spin Hamiltonian is necessary to reveal the detailed features.

Similar behavior was observed for the $\hbar\omega_1(\mathbf{q} \mp \mathbf{Q}, \pm \mathrm{QD})$ mode, as shown in Fig. 4b. In contrast with $\hbar\omega_2(\mathbf{q} \pm \mathbf{Q}, \pm \mathrm{QD})$, the linewidths are resolution limited, and the lifetime of the magnon is long at all the fields. At K and M points, the one-magnon energy of $\hbar\omega_2(\mathbf{q} \pm \mathbf{Q}, \pm \mathrm{QD})$ and the maximum of 2M-DoS were separated from one another. Hence, the repulsion was not strong enough for the one-magnon to avoid decay.

To survey the intensity change before and after the spectrum split, the magnetic field dependence of the intensity of the

one-magnon, remnant magnon, and their sum are shown in Fig. 4c for $\hbar\omega_2(\mathbf{q} \pm \mathbf{Q}, \pm \mathrm{QD})$ and 4d for $\hbar\omega_1(\mathbf{q} \mp \mathbf{Q}, \pm \mathrm{QD})$. The sums in the experiment were reproduced by the intensities of the calculated one-magnon indicated by the solid curves, and the sum rule was satisfied. The spectral transfer from the remnant magnon to one magnon was experimentally observed in the range of 3–5 T, which is far below the saturation field of 14 T[36]. Further theoretical study is required for the quantitative explanation.

Field dependence of magnon decay was also reported in pyrochlore Yb$_2$Ti$_2$O$_7$[37]. In contrast with RbFeCl$_3$, the magnon is fully decayed in the measured $\mathbf{q}-\hbar\omega$ range at 0 T due to strong quantum fluctuation, and a well-defined magnon appears at the energy calculated by linear spin-wave theory at the high field where the energy is out of two-magnon continuum. The one-magnon in RbFeCl$_3$ is well-defined at 0 T in the measured $\mathbf{q}-\hbar\omega$, and that of high-energy mode around $\Gamma$ point exhibits nontrivial field dependence. The behavior is entirely different from that of Yb$_2$Ti$_2$O$_7$.

In conclusion, using the INS technique and LESW, we investigated the magnetic excitations on RbFeCl$_3$ in the magnetic field. The observed excitations and their magnetic field dependences were semi-quantitatively explained by using LESW. Magnon decay was widely observed in the high-energy region where the longitudinal correlation was dominant and two-magnon density was large. Applying a magnetic field tunes the repulsive interaction between the one-magnon and two-magnon continuum, resulting in the quasiparticle avoiding decay in the high field, whereas the quasiparticle stays as it is in the low field. Our study demonstrates that the quasiparticle avoiding decay occurring under the strong interaction between the one-quasiparticle and continuum is ubiquitous, and the field control of quasiparticle decay is possible in the system, where the two-quasiparticle continuum covers a wide $\mathbf{q}-E$ space.

## Methods

Single crystals of RbFeCl$_3$ were grown using the vertical Bridgeman method[38]. The stoichiometric mixture of RbCl and FeCl$_2$ was dried for three days in a vacuum at 120 °C. The powder was sealed in an evacuated quartz tube set in a furnace at 650 °C and lowered at a rate of 3 mm/h. The obtained single crystals were characterized by magnetization measurement, transmission X-ray Laue diffraction, and cold neutron triple-axis spectrometer HER installed in JRR-3.

INS experiments at zero magnetic field were performed using a high-resolution chopper spectrometer (HRC)[39] cooperated by the High Energy Accelerator Research Organization (KEK) and the University of Tokyo at the materials and life science experimental facility of the Japan Proton Accelerator Research Complex. The measurement conditions were (i) the fixed incident neutron energies $E_i = 3.05$ and 5.09 meV at $T = 0.9$ K for $(h, k, 0)$ plane and (ii) $E_i = 10.2$ meV at $T = 1.6$ K for $(2h, -h, l)$ plane. The frequency of the Fermi chopper was 100 Hz, and the energy resolutions at the elastic line for $E_i = 3.05$, 5.09, and 10.2 meV were 0.072, 0.18, and 0.42 meV, respectively. We employed data collected at 100 K as the background for $E_i = 5.09$ and 10.2 meV.

The INS experiment in the magnetic field was performed using a hybrid spectrometer (HYSPEC) at the Spallation Neutron Source in Oak Ridge National Laboratory (ORNL)[40]. A $^3$He–$^4$He dilution cryostat was used to achieve a temperature of 0.1 K. The scattering plane was $(h, k, 0)$, and a magnetic field of up to 5 T was applied along the crystallographic $c$-axis. An incident neutron energy of 3.8 meV was selected using a Fermi chopper rotating at 180 Hz, resulting in an energy resolution of $\delta E = 0.14$ meV at the elastic line.

The integration ranges for inelastic neutron scattering spectra in Figs. 1a, 2a–f, and 3a, b are summarized in Table S1 in Supplementary Information. The instrumental resolution for the HYSPEC experiment was calculated using Monte Carlo neutron ray-tracing software MCViNE[41,42]. The details are described in Supplementary Information Section III.

## Data availability

The raw experimental data measured at the HRC spectrometer are stored on KEK's Neutron-Science Division computers. The raw experimental data measured at the HYSPEC spectrometer are stored on ORNL's Neutron-Scattering Division computers. The reduced experimental data and the theory data are available from the corresponding authors under simple requests.

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

## Acknowledgements

Prof. M. Matsumoto and Dr. R. Verresen are greatly appreciated for fruitful discussions. We are grateful to R. Ishii and H. Tanaka for the advice on growing single crystals and D. Kawana, T. Asami, R. Sugiura, and M. K. Graves-Brook for supporting us in the neutron scattering experiment at HRC and HYSPEC. The neutron experiment at the Materials and Life Science Experimental Facility of the J-PARC was performed using a user program (Proposal No. 2018S01). The neutron experiment at JRR-3 was performed using a user program (Proposal No. 21403).

A portion of this research used resources at the Spallation Neutron Source, a DOE Office of Science User Facility operated by the Oak Ridge National Laboratory. Travel expenses for the neutron scattering experiments performed using HYSPEC at ORNL, USA, were supported by the US-Japan Cooperative Research Program on Neutron Scattering. S. Hasegawa was supported by the Japan Society for the Promotion of Science through the Leading Graduate Schools (MERIT). This project was supported by JSPS KAKENHI Grant Nos. 19KK0069, 20K20896, and 21H04441.

## Author contributions

S.H., S.A., and T.M. carried out the experiment with the help of instrumental scientists B.W. and S.I. RbFeCl$_3$ single crystals were synthesized by S.H. The research framework was conceived by T.M. The data analysis was performed by S.H., B.W., G.S., and H.K. The text was written by S.H., H.K., Z.W., S.A., B.W., G.S., S.I., and T.M.

## Competing interests

The authors declare no competing interests.
