## [Peer Review File · Nature Communications]

Reviewers' Comments:

Reviewer #1:

Remarks to the Author:

Magnetic materials represent exemplar systems to study quasiparticle dynamics and therefore test many body theories. In their work, Hasegawa and co-authors study the magnon quasiparticles in a triangular lattice magnet and claim that applied magnetic fields act to "tune" the strength of interaction between single magnons and a multimagnon continuum. The field acts to tune the system from very weakly interacting to a strongly interacting regime. Both the magnon decays, that signify weak interactions, and magnon renormalizations/avoided crossings, that signify strong interactions, have been separately observed in various magnetic materials. The finding here is that both regimes can occur in a single material under the influence of a suitable tuning field. This is a potentially interesting finding, but it should be backed up with thorough analysis. This work comes close to presenting such an analysis, but more work is needed before I can recommend publication.

The authors first present zero field data, and an extended linear spin wave theory best fit that provides an adequate description of the data. Magnon-multi magnon interactions become apparent through an observed energy broadening of magnon modes upon application of a 2 T magnetic field. From the data in figure 2, it is clear that the linear spin wave theory no-longer provides an adequate description of the data, as expected when interactions become important. The authors then examine the line shapes of the magnon modes, fitting with a phenomenological Voigt function to extract the intrinsic energy width. Here I think the analysis could be made much stronger by properly accounting for the resolution of the instrument. This is especially important because the essential findings rely on an analysis of the magnon line shape. It will be especially helpful to understand the influence of the large out-of-plane dispersion here, since the neutron instrumentation used is highly focusing and will have a large out-of-plane divergence resulting in broad resolution along that direction.

The key findings are presented in figure 3 where the authors find "non-trivial spectrum splitting" associated with a deviation of the data from their model for applied magnetic fields above 2 T. They find that the highest energy single magnon mode sits at a lower energy than predicted by the linear theory, and that there is an additional mode at energies above the predicted magnon. The authors associated this additional mode with a "remnant magnon". They interpret the reduction in magnon energy to be a result of strong magnon - two magnon interactions in this field range. Again, this analysis would benefit from a more careful consideration and discussion of instrumental resolution effects. This analysis was particularly difficult to follow in the text because it relies on a deviation of the data from the linear spin wave prediction, but Fig. 3 (a) and (b) do not show any comparison between the linear spin wave prediction and the data, so the reader has to flip back and forth between fig 2 and fig 3 trying to match predicted modes with observed ones. The argument here could be made stronger if more direct comparisons of the data with the predictions from the extended linear spin wave model by including the model predictions, convolved with the instrumental resolution, on constant Q cuts in the main text.

Finally, the authors propose a simple model to understand their data, approximating the repulsive interaction between single and multi-magnons as a non-interacting two magnon density of states weighted difference in energy. This is a reasonable empirical approximate indicator and its evolution with the applied field seems to match some trends in the data, but on closer inspection is difficult to connect with the underlying physics. It would be helpful if the authors could comment/explain more about what is actually occurring in the cross-over from weak to strong interactions. Is it really the case that there is a critical value of interaction parameter to go from weak to strong coupling? Or finer field steps were taken, would a more smooth crossover occur? Can authors explain why the interaction effects are apparently maximized in this crossover region and why the 2ML signal decreases and apparent interaction between single magnon and continuum decreases for larger fields? 5T should still be far below the fields required to polarize this material. Addressing these questions in more detail would help strengthen the case for the reported findings here.

In addition to my comments above, I have a few other points for the authors to consider:

It would be helpful if the authors could more clearly indicate what distinguishes their findings from the field tuned magnon decays in Yb₂Ti₂O₇: Phys. Rev. Lett. 119, 057203

I don't understand the statement in the conclusion "We found that magnon avoiding decay... occurs regardless of the absence or presence of a continuum". This does not appear to be supported by the data or analysis. The authors do not show any detailed evidence in their data that a multi-magnon continuum is not in the regions of observed decay. Indeed the multi-magnon signal may have been too weak to observe with the statistics of this experiment. The analysis show a non-interacting 2 magnon density of states calculated from the linear spin wave model, and this density of states does not overlap the region of decay directly. However, it does not represent the actually multimagnon density of states that will be strongly renormalized by interactions. The authors should remove this statement or present more work to back it up.

Figure 3 references the SI in order to learn the meaning of the legends in Fig 3 making it very difficult for the reader to understand what is in those figures. It would be better to just describe the legend meaning directly in the figure caption.

Do the authors note any other features in their data, in addition to the reduction in mode energy, around the strong interaction regime ($\sim 3T$) that would be consistent with the avoided decay? For instance, can they show explicitly in their data that the ω_2 mode turns down to avoid the 2 magnon region around γ ?

The use of the term remnant magnon and symbol 2MH/L is confusing, since "remnant" implies the mode should be interpreted as a small contribution from a single magnon, while the symbol 2M seems to imply it should be interpreted as a two-magnon signal.

Reviewer #3:

Remarks to the Author:

In this manuscript, the authors study the interplay of magnon decay and spectrum renormalization in a quasi-one-dimensional antiferromagnet RbFeCl₃. They find that the strength of interaction between the two-magnon continuum and the one-magnon excitations changes with applied field, allowing them to tune from the weakly coupled regime (where spontaneous decay occurs) to the strongly coupled regime (where the one-magnon excitation is expelled from the continuum, avoiding decay). The authors then speculate on some application of these results to the transport of the magnons.

In my view, there are two key points of the paper:

- 1) At low fields the high-energy magnon excitations exhibit quasi-particle decay.
- 2) At high fields these decaying magnons are split, with spectral weight being expelled from the continuum (avoid the decay).

I think the authors have made a convincing case that this is indeed happening in RbFeCl₃. The neutron scattering data (the line cuts in particular) show clear evidence of this, with the high-lying branches first broadening (significantly enough to be visible in the color maps) and then sharpening and being renormalized down and away from the bulk of the two-magnon continuum. This is cemented by theoretical modelling (relying on a somewhat unconventional form of spin-wave theory) which matches the experiment well and semi-quantitatively describes the one-magnon excitations. The discrepancies between theory and experiment that appear at high field support the picture above -- that the repulsion from the two-magnon states [not captured at this linear level] is ramping up at high field and suppressing decay. This is further evidenced in the presence of two distinct peaks in the high field limit -- one being the 'expelled' magnon and the other a 'shadow' remaining in the continuum.

The observation of this physics is noteworthy -- though there are prior examples in a variety of systems (such as in Helium and in $\text{Ba}_3\text{CoSb}_2\text{O}_9$, as discussed by the authors) the field tunability here is novel.

One deficiency of the work in my view is that the more quantitative comparisons of the effects of decay were mostly ad-hoc -- e.g. using the two-magnon DOS as a proxy for the strength of one to two magnon scattering. While this is not an uncommon approximation given the complexity of these calculations, matrix element effects can be important and a more fulsome treatment would have been welcome. A comparison to more phenomenological models, such as that presented in *Nature Physics* 15, 750–753 (2019) would be appropriate (e.g. some of the non-trivial statements about the remnant spectral weight in the strong-coupling limit).

I think the cursory discussion of the effects on spin transport seem out of place as well. Almost no detail is given on what the authors are getting at here, save a sentence or two at the end of the paper. Given these are high energy magnons, and given that at temperatures high enough to populate these states (and thus render their lifetimes/decay relevant) the magnon description is likely not reliable, it is unclear if there is any relevance at all for spin current/transport. I would strongly suggest the authors revise or remove these comments.

Overall my impression of the work is quite positive. I think it certainly rises to the significance level appropriate for *Nature Communications* and would appeal to its broad readership. I do think some of the text could be improved to better communicate the core message (this is done in the introduction/abstract, but becomes a bit muddled when the data and theory itself are discussed). I would thus recommend publication in *Nature Communications*, with some minor comments/questions for the authors to consider below.

Questions/comments:

- 1) There are a number of typos in the paper. For example: i) Fig 4 Caption: "sume" -> "sum", ii) Line 240 "unlike a" (remove "a"), iii) Line 97 of the SM has a broken reference (appearing as ??). I would strongly suggest the authors do a thorough read of the paper to catch any further issues.
- 2) The authors state that conventional spin-wave theory fails to accurately describe the physics of this material. Have similar approaches tailored to the $S=1$ nature of this system (rather than $S=\infty$ in SWT) been considered? I.e. some of the $SU(3)$ -style or "flavor-wave" approaches?
- 3) The effect of hybridization between the one- and two-magnon states can depend significantly on the dimensionality of the system. I.e. in 2D as the magnon enters the continuum it always is expelled from the continuum (due to a log-divergence in the correction near the edge), while in 3D it does not. Here the system is 3D but with clear 1D and 2D sub-structures. Can the authors comment if this can potentially explain some of their observations? (e.g. the strength or tunability of the repulsion perhaps?)
- 4) In the supplemental material the authors state that the computed ordered moment from the theoretical calculations is significantly different than what is observed experimentally. They offer an explanation that the parameters should be renormalized away from the QCP to fix this by changing the ratio J_{eff}/D . Would such a change significantly modify the physics presented in the main text? Or would you expect these changes to be counteracted by additional renormalization of the spectrum by non-linear effects?

Responses to Reviewer's Comments

We are profoundly grateful for the insightful and constructive comments provided by the reviewers. Detailed responses to each comment have been delineated below for thorough clarification. Reviewer comments are presented in italic font, and our responses are in standard text.

Reviewer #1 (Remarks to the Author):

Magnetic materials represent exemplar systems to study quasiparticle dynamics and therefore test many body theories. In their work, Hasegawa and co-authors study the magnon quasiparticles in a triangular lattice magnet and claim that applied magnetic fields act to “tune” the strength of interaction between single magnons and a multimagnon continuum. The field acts to tune the system from very weakly interacting to a strongly interacting regime. Both the magnon decays, that signify weak interactions, and magnon renormalizations/avoided crossings, that signify strong interactions, have been separately observed in various magnetic materials. The finding here is that both regimes can occur in a single material under the influence of a suitable tuning field. This is a potentially interesting finding, but it should be backed up with thorough analysis. This work comes close to presenting such an analysis, but more work is needed before I can recommend publication.

*The authors first present zero field data, and an extended linear spin wave theory best fit that provides an adequate description of the data. Magnon-multi magnon interactions become apparent through an observed energy broadening of magnon modes upon application of a 2 T magnetic field. From the data in figure 2, it is clear that the linear spin wave theory no longer provides an adequate description of the data, as expected when interactions become important. The authors then examine the line shapes of the magnon modes, fitting with a phenomenological Voigt function to extract the intrinsic energy width. **Here I think the analysis could be made much stronger by properly accounting for the resolution of the instrument.** This is especially important because the essential findings rely on an analysis of the magnon line shape. **It will be especially helpful to understand the influence of the large out-of-plane dispersion here, since the neutron instrumentation used is highly focusing and will have a large out-of-plane divergence resulting in broad resolution along that direction.***

The key findings are presented in figure 3 where the authors find “non-trivial spectrum splitting” associated with a deviation of the data from their model for applied magnetic fields

above $2T$. They find that the highest energy single magnon mode sits at a lower energy than predicted by the linear theory, and that there is an additional mode at energies above the predicted magnon. The authors associated this additional mode with a “remnant magnon”. They interpret the reduction in magnon energy to be a result of strong magnon - two magnon interactions in this field range. *Again, this analysis would benefit from a more careful consideration and discussion of instrumental resolution effects. This analysis was particularly difficult to follow in the text because it relies on a deviation of the data from the linear spin wave prediction, but Fig. 3 (a) and (b) do not show any comparison between the linear spin wave prediction and the data, so the reader has to flip back and forth between fig 2 and fig 3 trying to match predicted modes with observed ones. The argument here could be made stronger if more direct comparisons of the data with the predictions from the extended linear spin wave model by including the model predictions, convolved with the instrumental resolution, on constant Q cuts in the main text.*

Firstly, we would like to apologize for the incorrect information regarding the integration ranges presented in Table S1 in the supplementary information. The correct integration range for c^* direction in Figs. 3a, 3b, and S1(a) should be 0.2 r.l.u instead of 0.01 r.l.u. The correct ranges for the $2a^*-b^*$ and b^* directions should be 0.04 r.l.u., not 0.02 r.l.u. These were simply mistakes, and they have been corrected.

Valuing the insightful suggestion from the reviewer, we redid our analysis to provide a more accurate depiction of the instrumental resolution. The calculation of the instrumental resolution was conducted utilizing the Monte Carlo neutron ray-tracing software, MCViNE. Dr. Gabriele Sala, who generously lent their expertise to this analysis, has been acknowledged through the addition as a coauthor in this manuscript. The recalculated INS spectra, obtained using the Linear Extended Spin Wave (LESW) theory and subsequently convoluted with the instrumental resolution, are displayed in Figs. 2g-2l.

Since the eigenenergies calculated by LESW do not perfectly reproduce the experimental data, we cannot use the calculated spectra convoluted by the instrumental resolution to discuss the detailed data. We, thus, need to seek an alternate analytical approach. In Figs. S7(a)-S7(c), constant \mathbf{q} cuts of the calculated spectra convoluted by the instrumental resolution are shown by the symbols. Please note that they are not experimental data, but the convoluted calculated spectra. The solid curves are the fitting results by using Gaussian, where the peak energies are the eigenenergies calculated by LESW. Since the fit to the data is reasonable, it is safe to assume that the instrumental resolution function is approximately Gaussian. $\hbar\omega$ dependence of Full Width at Half Maximum (FWHM) of the Gaussian, denoted as FWHM_G , for each mode is indicated by the symbols in Figs. S7(d)-S7(f).

FWHM_Gs at the energies of the observed excitations in Figs. 3a, 3b, and S1(a) are estimated from the interpolation of the data in Figs. S7(d)-S7(f). The measured constant q cuts are, thus, reanalyzed by Lorentzian convoluted by Gaussian, i.e., Voigt function, with the width of FWHM_G as shown in Figs. 3a, 3b, and S1(a). The instrumental resolutions are indicated by the horizontal bars. The work was conducted by Dr. Hodaka Kikuchi, who was later added as a collaborator on this paper. Details of the convolution analysis are explained in Section III, entitled Instrumental Resolution, in the supplementary information.

The results of the reanalysis align closely with our initial analysis. This is because the integration range in the c^* direction is wide, and the instrumental resolution effect is already incorporated in the spectra in Figs. 3a, 3b, and S1(a). Our pivotal findings, including non-trivial spectrum splitting associated with a deviation of the data from our model for applied magnetic fields above 2T, as stated in the initial manuscript, remain unaffected.

To improve readability, Figs. 3a, 3b, and S1(a) have been revised; the peaks are now indicated by colored curves, which are the fitted Voigt functions. These colors represent the type of mode calculated by LESW, eliminating the need for readers to flip back and forth between Fig 2 and Fig 3 to match predicted modes with observed ones. The meanings of these colors are explained in the legends of Figs. 3a and S1(a).

Finally, the authors propose a simple model to understand their data, approximating the repulsive interaction between single and multi-magnons as a non-interacting two magnon density of states weighted difference in energy. This is a reasonable empirical approximate indicator and its evolution with the applied field seems to match some trends in the data, but on closer inspection is difficult to connect with the underlying physics. It would be helpful if the authors could comment/explain more about what is actually occurring in the cross-over from weak to strong interactions. Is it really the case that there is a critical value of interaction parameter to go from weak to strong coupling? Or finer field steps were taken, would a more smooth crossover occur?

The change of the spectrum from the weak interaction case to the strong interaction case is drastic in the experiment. Dr. Ruben Verresen, one of the authors in the phenomenological model [9], suggested that the change is cross-over like in one or two dimension during a private communication. As shown in left top panel in Fig. L1, there is a long-lived quasiparticle outside the continuum, and it exists in all the k range though the intensity is small at small k . The weight is gradually shifted from the continuum onto the separate long-lived mode with the increase of the interaction, and the spectrum in strong interaction is shown in the bottom left panel. In contrast in three dimension the change is phase-

transition like; there is a threshold where the long-lived mode appears. This corresponds to the right column in Fig. L1. Even though RbFeCl₃ is weakly coupled ferromagnetic chains, the system is three dimension due to the interchain coupling, which leads to the drastic change in the spectrum. Meanwhile, the remnant magnon at $H \geq 3$ T would be due to the one-dimensional substructure. This discussion is based on the phenomenological theory, and further theoretical study considering the specific spin Hamiltonian is necessary to reveal the detailed features. This explanation is added in lines 200-209. Our gratitude to Dr. R. Verresen has been added to the acknowledgments.

Fig. L1 Avoided quasiparticle decay in a solvable model from Fig. 1 in Ref. [9].

Can authors explain why the interaction effects are apparently maximized in this crossover region and why the 2ML signal decreases and apparent interaction between single magnon and continuum decreases for larger fields? 5T should still be far below the fields required to polarize this material. Addressing these questions in more detail would help strengthen the case for the reported findings here.

The maximized interaction effects in the crossover region and the decrease of 2ML signal in the measured field region are the experimental facts, and they should be explained by future theoretical work. The explanation is added in lines 219-222.

It would be helpful if the authors could more clearly indicate what distinguishes their findings from the field tuned magnon decays in Yb₂Ti₂O₇: Phys. Rev. Lett. 119, 057203

As the reviewer suggested, the field dependence of magnon decay was reported also in pyrochlore YbTi_2O_7 [37]. In contrast with RbFeCl_3 , the magnon is fully decayed at 0 T in the measured $\mathbf{q}\text{-}\hbar\omega$ range due to strong quantum fluctuation, and a well-defined magnon appears at the energy calculated by linear spin-wave theory at high field where the energy is out of two-magnon continuum. The one-magnon in RbFeCl_3 is well-defined at 0 T in the measured $\mathbf{q}\text{-}\hbar\omega$ range, and that of high-energy mode around Γ point exhibits nontrivial field dependence. The behavior is entirely different from that of YbTi_2O_7 . The explanation is added in lines 223-229.

I don't understand the statement in the conclusion "We found that magnon avoiding decay ... occurs regardless of the absence or presence of a continuum". This does not appear to be supported by the data or analysis. The authors do not show any detailed evidence in their data that a multi-magnon continuum is not in the regions of observed decay. Indeed the multi-magnon signal may have been too weak to observe with the statistics of this experiment. The analysis show a non-interacting 2 magnon density of states calculated from the linear spin wave model, and this density of states does not overlap the region of decay directly. However, it does not represent the actually multimagnon density of states that will be strongly renormalized by interactions. The authors should remove this statement or present more work to back it up.

According to the suggestion, the sentence and related paragraph were deleted.

Figure 3 references the SI in order to learn the meaning of the legends in Fig 3 making it very difficult for the reader to understand what is in those figures. It would be better to just describe the legend meaning directly in the figure caption.

According to the suggestion, the caption of Fig. 3 was revised.

Do the authors note any other features in their data, in addition to the reduction in mode energy, around the strong interaction regime ($\sim 3T$) that would be consistent with the avoided decay? For instance, can they show explicitly in their data that the omega2 mode turns down to avoid the 2 magnon region around gamma?

I believe that "*omega2 mode*" is simply a typo of omega1 mode. The omega 2 mode indicated by red squares in Fig. 3d clearly turns down at $H \geq 3 T$. The omega 1 mode indicated by light

blue squares also turn down, but it is slight.

The use of the term remnant magnon and symbol $2M_{H/L}$ is confusing, since “remnant” implies the mode should be interpreted as a small contribution from a single magnon, while the symbol $2M$ seems to imply it should be interpreted as a two-magnon signal.

According to the suggestion, the terms $2M_L$ and $2M_H$ are changed to RM_H and RM_L , respectively.

Reviewer #3 (Remarks to the Author):

In this manuscript, the authors study the interplay of magnon decay and spectrum renormalization in a quasi-one-dimensional antiferromagnet $RbFeCl_3$. They find that the strength of interaction between the two-magnon continuum and the one-magnon excitations changes with applied field, allowing them to tune from the weakly coupled regime (where spontaneous decay occurs) to the strongly coupled regime (where the one-magnon excitation is expelled from the continuum, avoiding decay). The authors then speculate on some application of these results to the transport of the magnons.

In my view, there are two key points of the paper:

1) At low fields the high-energy magnon excitations exhibit quasi-particle decay.

2) At high fields these decaying magnons are split, with spectral weight being expelled from the continuum (avoid the decay).

I think the authors have made a convincing case that this is indeed happening in $RbFeCl_3$. The neutron scattering data (the line cuts in particular) show clear evidence of this, with the high-lying branches first broadening (significantly enough to be visible in the color maps) and then sharpening and being renormalized down and away from the bulk of the two-magnon continuum. This is cemented by theoretical modelling (relying on a somewhat unconventional form of spin-wave theory) which matches the experiment well and semi-quantitatively describes the one-magnon excitations. The discrepancies between theory and experiment that appear at high field support the picture above -- that the repulsion from the two-magnon states [not captured at this linear level] is ramping up at high field and suppressing decay. This is further evidenced in the presence of two distinct peaks in the high field limit -- one

being the 'expelled' magnon and the other a 'shadow' remaining in the continuum.

The observation of this physics is noteworthy -- though there are prior examples in a variety of systems (such as in Helium and in $Ba_3CoSb_2O_9$, as discussed by the authors) the field tunability here is novel.

One deficiency of the work in my view is that the more quantitative comparisons of the effects of decay were mostly ad-hoc -- e.g. using the two-magnon DOS as a proxy for the strength of one to two magnon scattering. While this is not an uncommon approximation given the complexity of these calculations, matrix element effects can be important and a more fulsome treatment would have been welcome. A comparison to more phenomenological models, such as that presented in *Nature Physics* 15, 750–753 (2019) would be been appropriate (e.g. some of the non-trivial statements about the remnant spectral weight in the strong-coupling limit).

I think the cursory discussion of the effects on spin transport seem out of place as well. Almost no detail is given on what the authors are getting at here, save a sentence or two at the end of the paper. Given these are high energy magnons, and given that at temperatures high enough to populate these states (and thus render their lifetimes/decay relevant) the magnon description is likely not reliable, it is unclear if there is any relevance at all for spin current/transport. *I would strongly suggest the authors revise or remove these comments.*

According to the suggestion, the last two sentences in the conclusion were deleted.

Overall my impression of the work is quite positive. I think it certainly rises to the significance level appropriate for *Nature Communications* and would appeal to its broad readership. I do think some of the text could be improved to better communicate the core message (this is done in the introduction/abstract, but becomes a bit muddled when the data and theory itself are discussed). *I would thus recommend publication in Nature Communications, with some minor comments/questions for the authors to consider below.*

The responses to questions/comments are appended below.

Questions/comments:

- i) There are a number of typos in the paper. For example: i) Fig 4 Caption: "sume" -> "sum",
- ii) Line 240 "unlike a" (remove "a"),
- iii) Line 97 of the SM has a broken reference (appearing

as ??). I would strongly suggest the authors do a thorough read of the paper to catch any further issues.

We revise the specific points raised by the referee. In addition, we have thoroughly read the paper and corrected several typos.

2) *The authors state that conventional spin-wave theory fails to accurately describe the physics of this material. Have similar approaches tailored to the $S=1$ nature of this system (rather than $S=\infty$ in SWT) been considered? I.e. some of the $SU(3)$ -style or "flavor-wave" approaches?*

Extended spin wave theory is the same as a generalized $SU(3)$ spin-wave theory. We added the explanation in lines 76-77.

3) *The effect of hybridization between the one- and two-magnon states can depend significantly on the dimensionality of the system. I.e. in 2D as the magnon enters the continuum it always is expelled from the continuum (due to a log-divergence in the correction near the edge), while in 3D it does not. Here the system is 3D but with clear 1D and 2D substructures. Can the authors comment if this can potentially explain some of their observations? (e.g. the strength or tunability of the repulsion perhaps?)*

As described in the response to reviewer 1, the change of the spectrum from the weak interaction case to the strong interaction case is cross-over like in one or two dimension. The one magnon is expelled from the continuum, and there is always a long-lived quasiparticle outside of it, as shown in the left top panel in Fig. L1. The remnant magnon exists at the energy of bare one-magnon in the continuum. The weight is gradually shifted from the continuum onto the separate long-lived mode with the increase of the interaction, and the spectrum in strong interaction is shown in the bottom left panel. In contrast in three dimension the change is phase-transition like; there is a threshold where the long-lived mode appears. This corresponds to the right column in Fig. L1. Even though RbFeCl_3 is weakly coupled ferromagnetic chains, the system is three dimension due to the interchain coupling, which leads to the drastic change in the spectrum. Meanwhile, the remnant magnon at $H \geq 3$ T would be due to the one-dimensional substructure. This discussion is based on the phenomenological theory, and further theoretical study considering the specific spin Hamiltonian is necessary to reveal the detailed features. We added the discussion related to the dimension in lines 200-209.

4) *In the supplemental material the authors state that the computed ordered moment from the theoretical calculations is significantly different than what is observed experimentally. They offer an explanation that the parameters should be renormalized away from the QCP to fix this by changing the ratio J_{eff}/D . Would such a change significantly modify the physics presented in the main text? Or would you expect these changes to be counteracted by additional renormalization of the spectrum by non-linear effects?*

In the field control of magnon decay, the interaction between one-magnon and two-magnon continuum plays an important role, and the absolute values of the exchange constants are irrelevant. The underestimate of J_{eff} does not affect the discussion in the main text. We added the explanation in lines 83-86 in the supplementary information.

Reviewers' Comments:

Reviewer #1:

Remarks to the Author:

The authors have thoughtfully considered and addressed my concerns and considerably improved their manuscript. I particularly appreciate the effort to quantitatively consider resolution effects in their inelastic neutron scattering data. I am now happy to recommend publication of this work in Nature Communications.

Reviewer #3:

Remarks to the Author:

The authors have mostly addressed the (minor) comments and questions that were raised in my report. For the reasons outlined in my first report I thus recommend publication.

The response to the other two referees also appears satisfactory to my eye. In particular, one larger point related to resolution effects raised by Referee #1 have been handled via a much more thorough analysis of the experimental and theoretical results. The smaller, more minor comments raised by the first referee have, in my opinion, been addressed appropriately.

Dear Editor,

We would like to express our sincere appreciation to both reviewers for recommending our work for publication in Nature Communications. We note that they did not raise any criticism regarding our revised manuscript in their comments below. We are deeply grateful for their constructive feedback.

Sincerely yours,

Takatsugu Masuda

Reviewer #1 (Remarks to the Author):

The authors have thoughtfully considered and addressed my concerns and considerably improved their manuscript. I particularly appreciate the effort to quantitatively consider resolution effects in their inelastic neutron scattering data. I am now happy to recommend publication of this work in Nature Communications.

Reviewer #3 (Remarks to the Author):

The authors have mostly addressed the (minor) comments and questions that were raised in my report. For the reasons outlined in my first report I thus recommend publication.

The response to the other two referees also appears satisfactory to my eye. In particular, one larger point related to resolution effects raised by Referee #1 have been handled via a much more thorough analysis of the experimental and theoretical results. The smaller, more minor comments raised by the first referee have, in my opinion, been addressed appropriately.